# Potential Therapeutic Effects of *Bifidobacterium breve* MCC1274 on Alzheimer’s Disease Pathologies in *App^NL-G-F^* Mice

**DOI:** 10.3390/nu16040538

**Published:** 2024-02-15

**Authors:** Mona Abdelhamid, Cha-Gyun Jung, Chunyu Zhou, Rieko Inoue, Yuxin Chen, Yoshiki Sento, Hideki Hida, Makoto Michikawa

**Affiliations:** 1Department of Biochemistry, Graduate School of Medical Sciences, Nagoya City University, 1 Kawasumi, Mizuho-cho, Mizuho-ku, Nagoya 467-8601, Japan; monaahmed92@vet.bsu.edu.eg (M.A.); haruu5916@gmail.com (C.Z.); rieinoue@med.nagoya-cu.ac.jp (R.I.); cyxbobo@gmail.com (Y.C.); 2Department of Neurophysiology and Brain Science, Graduate School of Medical Sciences, Nagoya City University, 1 Kawasumi, Mizuho-cho, Mizuho-ku, Nagoya 467-8601, Japan; hhida@med.nagoya-cu.ac.jp; 3Department of Anesthesiology and Intensive Care Medicine, Graduate School of Medical Sciences, Nagoya City University, 1 Kawasumi, Mizuho-cho, Mizuho-ku, Nagoya 467-8601, Japan; sentoyoshiki@yahoo.co.jp; 4Department of Geriatric Medicine School of Life, Dentistry at Niigata, Nippon Dental University, 1-8 Hamaura-cho, Chuo-ku, Niigata 951-8580, Japan

**Keywords:** Alzheimer’s disease, *Bifidobacterium breve* MCC1274, cognitive dysfunction, extracellular signal-regulated kinase, c-Jun *N*-terminal kinase, tau phosphorylation, glial activation, synapses

## Abstract

We previously demonstrated that orally supplemented *Bifidobacterium breve* MCC1274 (*B. breve* MCC1274) mitigated Alzheimer’s disease (AD) pathologies in both 7-month-old *App^NL-G-F^* mice and wild-type mice; thus, *B. breve* MCC1274 supplementation might potentially prevent the progression of AD. However, the possibility of using this probiotic as a treatment for AD remains unclear. Thus, we investigated the potential therapeutic effects of this probiotic on AD using 17-month-old *App^NL-G-F^* mice with memory deficits and amyloid beta saturation in the brain. *B. breve* MCC1274 supplementation ameliorated memory impairment via an amyloid-cascade-independent pathway. It reduced hippocampal and cortical levels of phosphorylated extracellular signal-regulated kinase and c-Jun *N*-terminal kinase as well as heat shock protein 90, which might have suppressed tau hyperphosphorylation and chronic stress. Moreover, *B. breve* MCC1274 supplementation increased hippocampal synaptic protein levels and upregulated neuronal activity. Thus, *B. breve* MCC1274 supplementation may alleviate cognitive dysfunction by reducing chronic stress and tau hyperphosphorylation, thereby enhancing both synaptic density and neuronal activity in 17-month-old *App^NL-G-F^* mice. Overall, this study suggests that *B. breve* MCC1274 has anti-AD effects and can be used as a potential treatment for AD.

## 1. Introduction

Alzheimer’s disease (AD) is a major cause of dementia and involves different pathogenic mechanisms that impair mental functions, especially cognition [1]. AD is mainly associated with a decline in explicit memory. AD pathogenesis involves pathogenic contributions from multiple components, such as amyloid beta (Aβ), hyperphosphorylated tau, and glial cell. Impaired Aβ clearance and/or an increase in its production leads to Aβ accumulation. Also, tau pathology propagates during AD progression. Aβ aggregates, together with tau accumulation, can cause glial activation and the subsequent neuroinflammation [2]. Several studies have suggested that Aβ aggregates critically contribute to the hyperphosphorylation of tau protein, which results in neuronal dysfunction and death, leading to cognitive dysfunction [3,4,5,6]. Tau protein plays an essential role in maintaining microtubule structure and promoting its assembly. In AD, hyperphosphorylation of tau protein inhibits its function, leading to the breakdown of microtubules, and the activity of tau protein depends on the degree of its hyperphosphorylation [7,8]. The number of patients with AD doubles every five years, and researchers continue to develop new treatments, including drugs that decrease the progression or treat symptoms of the disease. 

Probiotics have been shown to ameliorate cognitive dysfunction and delay the progression of AD by reducing oxidative stress and/or neuroinflammation [9,10]. Moreover, probiotics have been shown to enhance the neurotrophic factor in the brain that facilitates the differentiation and survival of neurons [11]. Furthermore, previous studies have suggested that oral supplementation of the probiotic *Bifidobacterium breve* MCC1274 (*B. breve* MCC1274) can mitigate AD pathologies in both AD model and wild-type (WT) mice by reducing Aβ production and neuroinflammation as well as improve cognitive deficits in individuals with mild cognitive impairment (MCI) [10,12,13,14]. Also, probiotics promote an increase in beneficial gut microbiota and their fermentation metabolites [15]. The gut microbiota plays a vital role in the regulation of neurological disorders because it regulates the gut–brain axis and has the ability to affect gut function and synthesis, which in turn affect learning and cognitive functions [16]. Bidirectional gut–brain axis communication is primarily achieved through endocrine, metabolic, and immune pathways or directly through the vagus nerve [17]. Probiotics increase the production of some neurotransmitters, activate the formation of short-chain fatty acids, or modulate the production of inflammatory cytokines [18,19,20]. 

Chronic stress causes neuronal atrophy, negatively affects the neurotransmitter system, and thus promotes cognitive decline; therefore, it plays a vital role in accelerating the progression of AD. Chronic stress has also been reported to accelerate tau hyperphosphorylation and memory impairment in WT and PS19 mice [21,22]. Furthermore, chronic stress enhances the deposition of Aβ plaques by increasing the levels of neurotoxic Aβ42 in female 5XFAD mice or raising Aβ42/Aβ40 ratios in APP-PS1 mice [23,24,25]. However, the mechanisms involved in the role of chronic stress in the accelerated progression of AD are not well understood. Previous studies have shown that cellular exposure to any chronic stress condition, including pathological conditions such as tissue damage, inflammation, oxidative stress, and heat shock, causes an increase in the expression of heat shock proteins (HSPs), which participate in the folding and assembly of newly synthesized proteins [26,27]. c-Jun *N*-terminal kinase (JNK) is a critical kinase that activates chronic stress and may cause neurological disorders [28]. Since JNK promotes phosphorylation of glucocorticoid receptors and inhibits their transcriptional activity, it also induces glucocorticoid maladaptive influence on the brain [29,30]. The activation of JNK is also reported during neuronal degeneration [31]. Moreover, chronic stress induces the persistent extracellular signal-regulated kinase (ERK) activation pathway, which can mediate cortical abnormalities [32,33]. 

Some studies have suggested that tau hyperphosphorylation is the main driver of cognitive dysfunction in patients with neurological disorders [34,35]. Different sites of tau hyperphosphorylation have different pathogenic effects; hyperphosphorylation of tau at Ser202/Thr205 sites sequesters normal microtubule-associated proteins from microtubules, while its binding to microtubules is inhibited by hyperphosphorylation at the Thr231 site [36,37]. Abnormally phosphorylated tau in the AD-affected brain is regulated by different protein kinases, such as JNK, ERK, cAMP-dependent protein kinase, and cyclin-dependent kinase 5 [38,39,40]. Furthermore, HSP90 plays a critical role in controlling tau hyperphosphorylation; its inhibition reduces hyperphosphorylation of tau through the alteration of some kinase activities and the stability of hyperphosphorylated tau [41,42,43].

Many studies have revealed that synaptic loss throughout brain regions is mainly correlated to cognitive dysfunction and appears in the early stages of AD [44,45]. The functions of synaptic proteins, particularly synaptophysin (SYP) and post-synaptic density 95 (PSD95), are crucial. SYP, a presynaptic protein, functions as a proxy for synaptic density [46], and PSD-95, a postsynaptic protein, is required for synaptic excitability [47]. Chronic stress has been shown to reduce the number of dendritic spines and the levels of synaptic proteins, and these behavioral and synaptic deficits were reversed by glutamate *N*-methyl-D-aspartate receptor antagonists [48]. c-Fos is expressed within neurons and can be used as a marker for neuronal activation [49,50]. Further, c-Fos expression is essential for enhancing memory formation [51,52]. 

Many recent studies have demonstrated the benefits of probiotics on cognitive function. Probiotic supplementation was associated with better cognitive function in MCI older adults and benefited sleep quality in older adults [53,54]. In addition, oral administration of *Bifidobacterium lactis* ameliorates cognitive dysfunction in Aβ-injected mice through attenuation of ERK and JNK phosphorylation [55]. As well as it has been reported that *Lactobacillus* species act as protective from environmental stressors [56], and *Bifidobacterium breve* alleviated Aβ-induced cognitive impairment and enhanced synaptic function in AD mice [57]. We previously demonstrated that orally supplemented *B. breve* MCC1274 mitigated AD pathologies, as indicated by improved memory impairment, reduced amyloidosis and neuroinflammation, and improved synaptic plasticity, in both young AD model and WT mice [10,12,13]. Furthermore, this probiotic improved memory function in individuals with MCI [14]. These studies support the use of *B. breve* MCC1274 supplementation as a potential therapeutic strategy for preventing the progression of AD. However, whether this probiotic can be used as a curative treatment for AD remains unknown. In the present study, 13-month-old amyloid precursor protein (APP) knock-in (KI) (*App^NL-G-F^*) mice were administered *B. breve* MCC1274 (1 × 10^9^ cfu/mouse) for four months via oral gavage. We then demonstrated the effects of this probiotic on cognitive dysfunction, Aβ production, glial activation, chronic stress marker levels, synaptic and hyperphosphorylated tau protein levels, and neuronal activity.

## 2. Materials and Methods

### 2.1. Animal Model and Probiotic Supplementation

APP-KI (*App^NL-G-F^*) mice, carrying Swedish, Beyreuther/Iberian, and Arctic mutations, in which Aβ production and deposition were overexpressed without changes in APP expression levels or even its metabolic products and showed cognitive dysfunction, as well as neuroinflammation [58], were provided by the RIKEN BRC through the National Bio-Resource Project of MEXT, Japan. All experiments were performed in accordance with the National Institute of Health Guide for the Care and Use of Laboratory Animals. The transgenic mice used in the experiments were aged 13 months. The mice were administered *B. breve* MCC1274 (1 × 10^9^ cfu/mouse/day) via oral gavage five times per week for four months (*n* = 17). Another group of mice that received saline was considered the control group (*n* = 16). *B. breve* MCC1274 was prepared as described in a previous study [13], isolated from infant feces, grown in a glucose- and yeast-rich medium, and collected by centrifugation. All animals had access to water and food ad libitum, housed under a 12-h dark/light cycle, and were weighed every two weeks during the experimental period. We then evaluated episodic memory using a novel object recognition (NOR) test. Finally, all mice were deeply anesthetized using sevoflurane inhalation for extracting the brain, which was later used for immunofluorescence staining, western blotting analysis, or enzyme-linked immunosorbent assay (ELISA). 

### 2.2. Novel Object Recognition Test

A NOR test was conducted in 17-month-old *App^NL-G-F^* mice. This test is used to measure episodic memory and is based on the attraction of the animals toward the new object. The test was performed as previously described [59]. Briefly, the test relies on habituation, training, and test sessions. The sessions involve visual exploration of the box without any objects, with two identical objects (familiar objects), or with both familiar and novel objects for 5 min each. The habituation session was continued for three days, followed by a training session on the 4th day, then the test session occurred on the 5th day. Exploration times for all the objects were recorded using a video camera. We then measured the discrimination index (DI) that reflected the difference in the exploration time for the novel object and the familiar object as a proportion of the total exploration time for both objects. NOR analysis was performed in a blinded manner.

### 2.3. Samples Preparation

The experimental animals were perfused intracardially with phosphate-buffered saline (PBS). All brains were divided into two hemispheres, which were snap-frozen. One hemisphere was homogenized in 19 volumes of Tris-buffered saline (TBS) buffer. Homogenates were centrifuged at 100,000 rpm for 20 min at 4 °C, and the resultant supernatant was collected as the TBS soluble fraction. The pellet was homogenized with 10 volumes of 6 M guanidine HCl, incubated in the dark for 1 h at room temperature (RT), and centrifuged at 100,000 rpm for 20 min at 4 °C. The resulting supernatant was collected as the guanidine-soluble fraction. The other hemisphere was homogenized in 10 volumes of radio-immunoprecipitation assay (RIPA) buffer (50 mM Tris-HCl, 150 mM NaCl, 1% Nonidet P-40, 0.5% sodium deoxycholate, and 0.1% sodium dodecyl sulfate [SDS]; pH, 7.6) containing phosphatase and protease inhibitor cocktails (FUJIFILM Wako Pure Chemical Corporation, Osaka, Japan). The homogenates were centrifuged at 12,000 rpm for 30 min at 4 °C, and the supernatants were used for western blot analysis. 

### 2.4. ELISA

TBS and guanidine soluble fractions from cortical and hippocampal regions of the brains were used to quantify the levels of soluble and insoluble Aβ40 and Aβ42 using sandwich Aβ ELISA kits (FUJIFILM Wako Pure Chemical Corporation, Osaka, Japan), as previously described, according to the manufacturer’s instructions [10,12,59]. The samples were analyzed in duplicate and were parallel to the standard curve. Aβ levels were normalized to the weight of the brain tissue.

### 2.5. Immunofluorescence Staining

Euthanized mice were intracardially perfused with PBS, followed by 4% paraformaldehyde (PFA). The subsequently extracted brains were fixed in 4% PFA for three days and immersed in 30% sucrose solution for two days. Brain sections (thickness, 40 µm) were prepared using a vibratome (Leica Microsystems, Wetzlar, Germany) and kept in a cryoprotection solution (250 mM polyvinylpyrrolidone, 150 mM NaCl, 30% sucrose, and 0.1 M PBS) at −20 °C. For immunofluorescence staining, antigen retrieval was performed using citrate buffer, followed by blocking with 5% goat serum for 1 h. Subsequently, the sections were incubated overnight with anti-82E1 antibody and rabbit polyclonal anti-glial fibrillary acidic protein (GFAP) (1:100, Sigma-Aldrich, St. Louis, MO, USA) or rabbit polyclonal anti-Iba1 (1:200, FUJIFILM Wako Pure Chemical Corporation) antibody, followed by incubation for 1 h at RT with goat anti-mouse Alexa Fluor 488- and goat anti-rabbit Alexa Fluor 568-conjugated secondary antibodies (Thermo Fisher Scientific, Rockford, IL, USA). Nuclear morphology was observed using 4′,6-diamidino-2-phenylindole (DAPI; Thermo Fisher Scientific). All images were captured using a confocal fluorescence microscope (SpinSR10 (Olympus, Tokyo, Japan)). The plaque area was detected using the 82E1 antibody, and the number of GFAP- and Iba1-positive cells were quantified using ImageJ software (https://imagej.net/ij/index.html, accessed on 12 February 2024). To evaluate Aβ plaques, we quantified the immunoreactive areas divided by the total area. We also counted the number of GFAP- and Iba1-positive cells in four randomly selected fields of the cortex and two fields of the hippocampus per animal (1 × 1 mm^2^ per field). The average of data from at least three sections per animal was used as an individual value to reduce variance among brain sections. 

### 2.6. Western Blot Analysis 

Protein concentrations in RIPA-soluble fractions from both cortical and hippocampal homogenates were measured using the PierceTM BCA Protein Assay kit (Thermo Fisher Scientific), according to the manufacturer’s instructions. Twenty micrograms of protein per sample were separated using SDS-polyacrylamide gel electrophoresis (SDS-PAGE), transferred to Immobilon-P membranes (Millipore, Billerica, MA, USA), blocked with 5% skim milk in TBS buffer, probed overnight at 4 °C for the primary antibodies, including anti-Iba1 (019-19741, Wako), anti-GFAP (Sigma-Aldrich), anti-HSP90 (Cell Signaling, Danvers, MA, USA), anti-phospho-ERK (Cell Signaling), anti-ERK (Cell Signaling), anti-phospho-JNK (Cell Signaling), anti-JNK (Cell Signaling), anti-PSD95 (Cell signaling), anti-SYP (Abcam, Cambridge, UK), anti-c-Fos (EnCor Biotechnology, Gainesville, FL, USA), anti-total tau (Biolegend, San Diego, CA, USA), anti-phospho-tau (AT180, Invitrogen, Carlsbad, CA, USA), anti-phospho-tau (AT8, Thermo Fisher Scientific), and anti-actin (Proteintech Group, Tokyo, Japan) antibodies. Subsequently, the membranes were incubated for 1 h with species-specific HRP-conjugated secondary antibodies. Signals were enhanced using chemiluminescence Immunostar reagents (FUJIFILM Wako Pure Chemical Corporation), visualized using an Amersham Imager 680 (GE Healthcare, Marlborough, MA, USA), and quantified by ImageJ software. The integrated optical density data were normalized to β-actin levels and expressed as relative protein levels.

### 2.7. Statistical Analysis

All data are shown as mean ± standard deviation values. A two-tailed unpaired Student’s *t*-test was used to determine significant differences between the groups, considering that the data were statistically significant when the *p*-value was less than 0.05. GraphPad Prism 7.0 software (San Diego, CA, USA) was used for the statistical analyses. All experiments generated similar results, and the data were assumed to be normally distributed.

## 3. Results

### 3.1. Effect of B. breve MCC1274 Administration on Body Weight

First, we evaluated whether *B. breve* MCC1274 administration has an effect on the average body weight of 17-month-old *App^NL-G-F^* mice every two weeks for four months. No significant differences in the body weights of *B. breve* MCC1274-supplemented and control groups were observed during the supplementation period (Appendix A). This result indicates that *B. breve* MCC1274 supplementation did not have a significant effect on the body weights of aged *App^NL-G-F^* mice.

### 3.2. Effect of B. breve MCC1274 Administration on Novel Object Recognition 

The cognitive function of aged *App^NL-G-F^* mice was evaluated by the NOR test. In the training session, no significant difference in basal precognitive behavior between the saline and probiotic groups was observed (Figure 1A). However, 24 h after the training session, the *B. breve* MCC1274-supplemented group displayed significantly improved recognition in exploring new objects, as indicated by the significant increase in the total exploration time for the novel object compared with the familiar object (Figure 1B). The saline group did not show any difference in the total exploration time between the familiar and novel objects (Figure 1B), which meant that this group did not display any improvement in cognitive ability. Consistently, the discrimination index significantly increased in the *B. breve* MCC1274-supplemented group compared with the control group (Figure 1C). These data indicate that *B. breve* MCC1274 administration improved cognition dysfunction in aged *App^NL-G-F^* mice.

### 3.3. Effect of B. breve MCC1274 Administration on Aβ40 and Aβ42 Levels

As Aβ plays an essential role in memory loss and the progression of AD, we evaluated the effects of *B. breve* MCC1274 administration on Aβ levels in both the cortex and hippocampus using ELISA. Previously, we had demonstrated that *B. breve* MCC1274 administration reduced hippocampal Aβ production and deposition in 7-month-old *App^NL-G-F^* mice and reduced hippocampal soluble Aβ42 levels in WT mice of the same age [10,12]. In the present study, no significant difference in the levels of both soluble and insoluble Aβ40 and Aβ42 in both hippocampal and cortical homogenates was observed between the *B. breve* MCC1274-supplemented and saline groups (Figure 2A,B). Therefore, *B. breve* MCC1274 administration has no effect Aβ levels in aged *App^NL-G-F^* mice and may improve memory impairment induced via an amyloid-cascade-independent pathway. 

### 3.4. Effect of B. breve MCC1274 Administration on Glial Cells Cluster around Aβ Plaques 

In view of the dual role of glial cells in the brain, either in reducing Aβ-mediated neurotoxicity by improving its degradation and phagocytosis in normal cases or in inducing neuroinflammation by stimulating the release of inflammatory cytokines leading to progression of neurodegeneration [60,61] in patients with AD, we evaluated the impact of *B. breve* MCC1274 supplementation on microglial activation and astrogliosis in the brains of aged *App^NL-G-F^* mice. In previous studies, *B. breve* MCC1274 administration reduced hippocampal microglial activation but did not affect astrogliosis in 7-month-old *App^NL-G-F^* or WT mice [10,12]. In the present study, brain sections were stained using anti-82E1 (recognizing both Aβ40 and Aβ42) and either anti-Iba1 (a microglia marker) or anti-GFAP (an astrocytic marker) antibodies. Microglia (red) with short, thick, and dystrophic processes were closely associated with senile plaques (green) (Figure 3A,B). In addition, astrocytes (red) with hypertrophic processes were observed around senile plaques (green) (Figure 4A,B). The number of Iba1- and GFAP-immunoreactive glial cells and Aβ plaques in both cortical and hippocampal regions remained unchanged between the *B. breve* MCC1274-supplemented and control groups (Figure 3 and Figure 4). 

The results of immunostaining analysis were confirmed using western blotting; no significant differences in the cortical and hippocampal levels of both Iba1 and GFAP were observed between the two groups (Figure 5A,B). Both immunostaining and western blotting analyses confirmed that *B. breve* MCC1274 supplementation had no effect on microglial activation and astrogliosis in aged *App^NL-G-F^* mice and that neuroinflammation was not involved in the improvement in cognitive function caused by *B. breve* MCC1274 administration.

### 3.5. Effect of B. breve MCC1274 Administration on Chronic Stress 

Chronic stress has been reported to accelerate AD progression and elevate the levels of HSP90 and phosphorylated (p-) ERK and JNK [26,28,32]. Therefore, we measured the levels of these chronic stress markers in the brain tissues of aged *App^NL-G-F^* mice using western blotting. HSP90 and p-JNK were significantly downregulated in both the hippocampus and cortex in the *B. breve* MCC1274-supplemented group compared with the control group (Figure 6A,B). p-ERK was significantly downregulated only in the cortex, and a non-significant decrease in its level was observed in the hippocampus with the probiotic supplementation (Figure 6A,B). These findings indicate that *B. breve* MCC1274 administration reduces the effects of chronic stress on the brains of aged *App^NL-G-F^* mice by reducing the levels of kinases and HSP90.

### 3.6. Effect of B. breve MCC1274 Administration on Synaptic Protein Levels 

Considering the adverse effect of chronic stress on synapses, we evaluated both cortical and hippocampal levels of post- and pre-synaptic proteins (PSD95 and SYP, respectively) using western blotting in aged *App^NL-G-F^* mice. Chronic stress contributes to the loss of synaptic connections, as well as behavioral and synaptic deficits [48,62]. *B. breve* MCC1274 administration upregulated SYP and PSD95 in the hippocampus but not in the cortex (Figure 7A,B), consistent with the findings in 7-month-old *App^NL-G-F^* and WT mice in our previous study. In addition, we evaluated the hippocampal and cortical levels of c-Fos using western blotting to determine the effect of *B. breve* MCC1274 on activated neurons. We found that c-Fos was significantly upregulated in both cortical and hippocampal tissues in the *B. breve* MCC1274-supplemented group compared with the control group (Figure 7A,B). Thus, *B. breve* MCC1274 supplementation may improve synaptic dysfunction in aged *App^NL-G-F^* mice, probably by reducing the levels of chronic stress markers and increasing neuronal activity. 

### 3.7. Effect of B. breve MCC1274 Administration on Hyperphosphorylated Tau Levels

As mentioned above, *B. breve* MCC1274 administration reduced the levels of chronic stress markers, such as HSP90, p-ERK, and p-JNK, and improved synaptic plasticity through upregulation of PSD95 and SYP protein levels. Chronic stress increases tau hyperphosphorylation in synapses, leading to synaptic loss and dendritic remodeling [63]. Therefore, we evaluated the levels of both total and phosphorylated tau using western blotting. The findings showed that *B. breve* MCC1274 administration reduced phosphorylated tau level at Thr231 (AT180) and did not affect both total tau and phosphorylated tau levels at Ser202 and Thr205 (AT8) in both the hippocampus and cortex (Figure 8A,B). Thus, these findings suggest that *B. breve* MCC1274 administration reduces the levels of chronic stress markers, which probably leads to reduced levels of phosphorylated tau, thus improving synaptic plasticity in aged *App^NL-G-F^* mice. 

## 4. Discussion

In the current study, we evaluated the effects of *B. breve* MCC1274 supplementation on AD pathology in 17-month-old *App^NL-G-F^* mice, including its effects on recognition ability, amyloid production, neuroinflammation, chronic stress, synaptic plasticity, and tau phosphorylation. The present study provides new possibilities for the potential treatment of AD. The results showed that oral *B. breve* MCC1274 supplementation improved memory impairment and reduced chronic stress, which might be indications of reduced tau phosphorylation at Thr231 and improved synaptic plasticity, as well as neuronal activity.

*App^NL-G-F^* mice are widely used as standard mouse models for identifying AD pathologies, as these mice carry Swedish, Beyreuther/Iberian, and Arctic mutations that cause Aβ deposition, cognitive dysfunction, neuroinflammation, astrogliosis, and microgliosis in an age-dependent manner [58]. Therefore, in the present study, 13-month-old *App^NL-G-F^* mice were orally administered *B. breve* MCC1274 or saline for four months. The role of probiotics in ameliorating memory impairment has been attracting attention from different research groups. Supplementation of *L. casei, L. acidophilus, L. fermentum, B. bifidum, B. longum,* and *B. breve,* has been reported to positively affect cognitive function [10,64,65]. In the present study, *B. breve* MCC1274 administration significantly improved cognitive dysfunction in aged *App^NL-G-F^* mice, consistent with the findings of our previous study conducted among young *App^NL-G-F^* mice. Cognitive dysfunction, manifested as the loss of the ability to differentiate between familiar and novel objects in the NOR test, was observed in the saline-treated group compared with the probiotic-supplemented group. The probiotic-supplemented group could identify the novel object and spent more time exploring it than exploring the familiar object. Therefore, these findings and the previous ones strongly confirmed the vital role of *B. breve* MCC1274 administration in the amelioration of cognitive dysfunction in *App^NL-G-F^* mice.

Aβ is considered to be the initiator of the disease because its accumulation is usually associated with cognitive decline, neuroinflammation, tau hyperphosphorylation, and synaptic dysfunction [66]. Therefore, we examined whether *B. breve* MCC1274 administration affected Aβ levels in the brains of aged *App^NL-G-F^* mice. The results indicated that *B. breve* MCC1274 administration had no effect on Aβ deposition or soluble and insoluble Aβ levels in both cortical and hippocampal tissues of these mice. Thus, we conclude that the improvement in cognitive impairment caused by *B. breve* MCC1274 administration in 17-month-old *App^NL-G-F^* mice might be induced via an amyloid-cascade-independent pathway. Aβ deposition in the brain is associated with neuroinflammation; therefore, we examined whether *B. breve* MCC1274 administration affected astrogliosis and microglial activation in the brains of aged *App^NL-G-F^* mice. It has been reported that *B. breve* MCC1274 administration reduced both Aβ deposition and neuroinflammation in the hippocampus of young *App^NL-G-F^* and WT mice [10,12]. In the present study, the results of western blotting and immunofluorescence staining indicated that this probiotic could not reduce the protein levels and numbers of GFAP- and Iba1-positive cells caused by aging and Aβ deposition in the brains of aged *App^NL-G-F^* mice. Thus, these data indicate that *B. breve* MCC1274 administration has no effect on astrogliosis and microglial activation in 17-month-old *App^NL-G-F^* mice.

To understand the other benefits of *B. breve* MCC1274 administration, we examined whether *B. breve* MCC1274 administration affected the levels of chronic stress markers in the brains of aged *App^NL-G-F^* mice. Activation of the MAPK/ERK signaling pathway following exposure to stressors has been noted as a direct cellular response. In addition, activation of the JNK pathway and induction of HSPs are also reported as responses to exposure to stress [26,67,68,69]. In the present study, western blotting revealed that the probiotic-supplemented group had significantly lower cortical and hippocampal levels of both phosphorylated JNK and ERK1/2 and HSP90 protein levels in 17-month-old *App^NL-G-F^* mice, indicating that oral administration of *B. breve* MCC1274 reduced the levels of chronic stress markers in the brains of these mice. Therefore, these data indicate that *B. breve* MCC1274 administration can be used to reduce the levels of chronic stress markers in the brain. ERK plays an essential role in synaptic plasticity in the hippocampus, and thus, abnormal ERK activation contributes to cognitive dysfunction in AD patients. We found that *B. breve* MCC1274 administration in aged *App^NL-G-F^* mice reduced ERK activation. These results suggest that the effect of *B. breve* MCC1274 on ERK activation depends on the age of the mouse or the pathological condition of *App^NL-G-F^* mice. Additional studies are required to determine the mechanism by which *B. breve* MCC1274 supplementation alters ERK activation in 17-month-old *App^NL-G-F^* mice.

Several studies have examined the direct link between chronic stress and synaptic plasticity; chronic stress downregulates synaptic proteins and induces dendritic alterations of neurons in the brain, and the activation of the ERK pathway in the hippocampus influences learning and memory skills by reducing synaptic plasticity [70,71,72]. Moreover, we recently reported that *B. breve* MCC1274 administration upregulated pre- and post-synaptic protein levels in the hippocampi of young *App^NL-G-F^* and WT mice [10,12]. Similarly, in the present study, we showed that this probiotic significantly increased both PSD95 and SYP protein levels in the hippocampal tissues of aged *App^NL-G-F^* mice, confirming that *B. breve* MCC1274 administration upregulated synaptic plasticity. In addition, several studies have shown that increased neuronal activity plays a critical role in the upregulation of synaptic plasticity [73,74]. In the current study, we found that *B. breve* MCC1274 administration significantly upregulated neuronal activity, as indicated by the increased levels of c-Fos protein in both hippocampal and cortical tissues in the probiotic-supplemented group compared with the saline-treated group. Thus, the present study and our previous study confirmed the significant role of *B. breve* MCC1274 supplementation in improving synaptic plasticity and neuronal activity, which might have been caused by reduced chronic stress in 17-month-old *App^NL-G-F^* mice. 

Chronic stress has been reported to accelerate the progression of AD since it induces both abnormal tau hyperphosphorylation in the brain and cognitive deficits [21]. As mentioned above, p-ERK, p-JNK, and HSP90 are upregulated on exposure to stress and are considered markers of chronic stress. They also play a critical role in controlling tau hyperphosphorylation in the AD-affected brain and reducing the stability of the hyperphosphorylated tau [36,37,38,39,40]. The results of the present study revealed that oral *B. breve* MCC1274 administration reduced tau hyperphosphorylation at Thr231 in the hippocampi of aged *App^NL-G-F^* mice. Microtubule stability has been reported to be disrupted by hyperphosphorylation of tau at Thr231, leading to synaptic loss and neural degeneration, and is strongly associated with the accumulation of neurofibrillary tangles that impair memory skills [36,37,38]. In addition, tau hyperphosphorylation might be the main driver of cognitive dysfunction in neurological disorders. Thus, the findings strongly confirm the vital role of *B. breve* MCC1274 supplementation in reducing hyperphosphorylation of tau at Thr231, which might have resulted from reduced levels of chronic stress markers, and in improving memory impairment in 17-month-old *App^NL-G-F^* mice.

Chronic stress induces alterations in cognitive function; it considerably weakens and reduces connectivity between neurons due to the loss of dendrites and/or dendritic spines. It also inhibits the Ca^2+^-cAMP signaling pathway and increases voltage-gated potassium channel activity, leading to working memory and cognitive deficits. Furthermore, these deficits and dendritic spine loss were reported to be improved by treatment with kinase inhibitors [62]. The findings of the present study and the previous studies provide new insights into the vital role of reduced chronic stress, resulting from reduced levels of phosphorylated ERK and JNK and HSP90 caused by *B. breve* MCC1274 supplementation, in alleviating neuropathological alterations, which involve reducing tau hyperphosphorylation, increasing synaptic density, and increasing neuronal activation, thus ameliorating memory impairment. Another mechanism that could explain the interaction between brain function and probiotic supplementation involves the beneficial bacteria in the intestine that can modulate mood status and cognitive ability through the production of neurotransmitters and/or neurochemicals, such as γ-aminobutyric acid, serotonin, and dopamine, that reduce depression and stress-related behaviors [75,76]. Further studies are required to elucidate the possible mechanisms involved in alleviating neuropathological alterations by *B. breve* MCC1274. 

## 5. Conclusions

*B. breve* MCC1274 administration might exert an ameliorative effect on AD in 17-month-old *App^NL-G-F^* mice partly by reducing chronic stress via inhibiting tau hyperphosphorylation at Thr231 and improving synaptic density and neuronal activation, finally improving cognitive deficits. The present observations are consistent with the findings of our previous study that *B. breve* MCC1274 administration ameliorated memory impairment, increased synaptic plasticity, and inhibited tau hyperphosphorylation. However, *B*. *breve* MCC1274 supplementation did not have any beneficial effects on neuroinflammation and Aβ levels in aged *App^NL-G-F^* mice, contrary to our previous findings in young *App^NL-G-F^* and WT mice. Overall, the findings of the present study and previous observations suggest that *B*. *breve* MCC1274 administration has anti-AD effects and can be used as a treatment for AD. 

## Figures and Tables

**Figure 1 nutrients-16-00538-f001:**
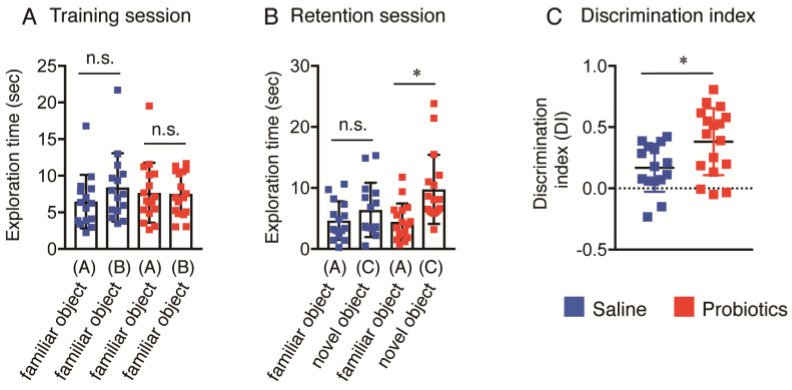
Oral *B. breve* MCC1274 administration ameliorates cognition dysfunction in 17-month-old *App^NL-G-F^* mice. (**A**) Object exploration during the training session. The saline and probiotic groups showed similar exploration times for both familiar objects. (**B**) Object exploration during the retention session. The saline group showed no difference in exploration time between familiar and novel objects. The probiotic group had a longer exploration time for the novel object than for the familiar object. (**C**) Discrimination index (DI). The relative preference for the novel object was calculated using the DI. Data are expressed as the mean ± standard deviation (*n* = 16−17 in each group). * *p* < 0.05, compared with the saline group; n.s., no significant difference; data were analyzed using Student’s *t*-test.

**Figure 2 nutrients-16-00538-f002:**
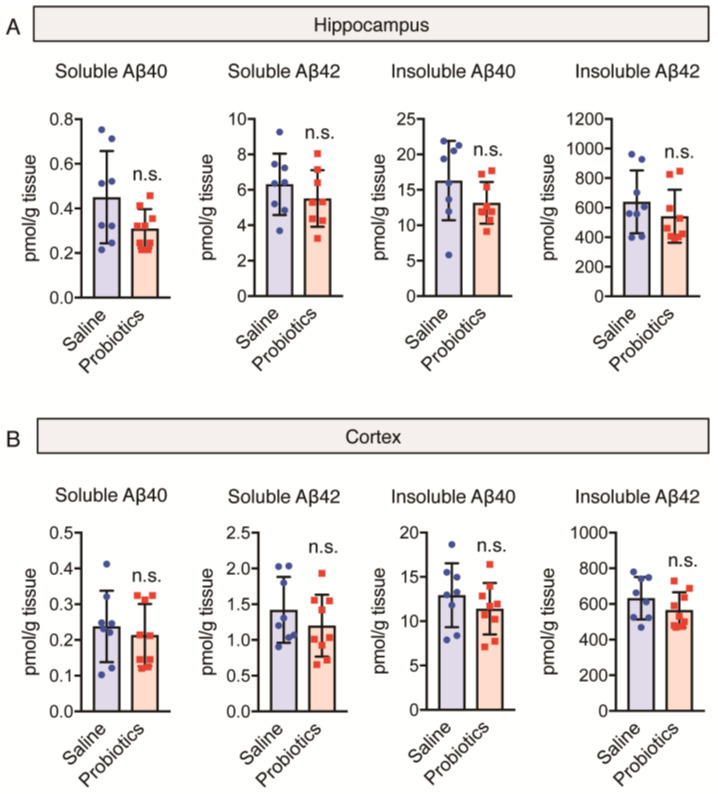
Oral *B. breve* MCC1274 administration does not alter amyloid beta levels in the brains of 17-month-old *App^NL-G-F^* mice. Levels of soluble Aβ40 and Aβ42, as well as insoluble Aβ40 and Aβ42 in hippocampal (**A**) and cortical (**B**) tissues, were measured using sandwich enzyme-linked immunosorbent assay. Aβ levels were normalized to the brain tissue weight. Data are represented as the mean ± standard deviation (*n* = 8–9 in each group). n.s., no significant difference; data were analyzed using Student’s *t*-test.

**Figure 3 nutrients-16-00538-f003:**
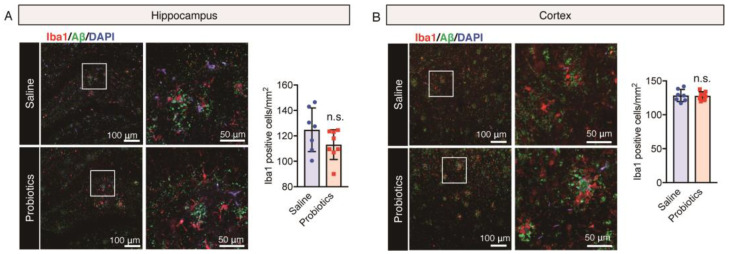
Oral *B. breve* MCC1274 administration does not change microglia clusters around Aβ plaques in the brains of 17-month-old *App^NL-G-F^* mice. Brain sections were stained using anti-82E1 antibody, which detected Aβ plaques (green), and anti-microglial Iba1 (red) antibodies, as well as DAPI that stained cell nuclei (blue). Representative immunofluorescence images of the hippocampus (**A**) and cortex (**B**) and quantitative analyses of Aβ plaques with Iba1. Data are represented as the mean ± standard deviation (*n* = 7 in each group). n.s., no significant difference; data were analyzed using Student’s *t*-test.

**Figure 4 nutrients-16-00538-f004:**
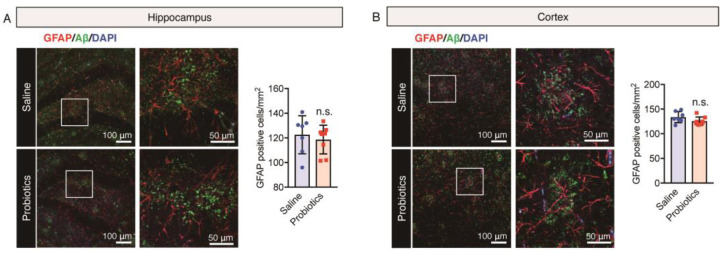
Oral *B. breve* MCC1274 administration does not change astrocyte clusters around Aβ plaques in the brains of 17-month-old *App^NL-G-F^* mice. Brain sections were stained using anti-82E1 antibody, which detected Aβ plaques (green), and anti-astrocytic glial fibrillary acidic protein (GFAP, red) antibodies, as well as DAPI that stained cell nuclei (blue). Representative immunofluorescence images of the hippocampus (**A**) and cortex (**B**) and quantitative analyses of Aβ plaques with GFAP. Data are represented as the mean ± standard deviation (*n* = 7 in each group). n.s., no significant difference; data were analyzed using Student’s *t*-test.

**Figure 5 nutrients-16-00538-f005:**
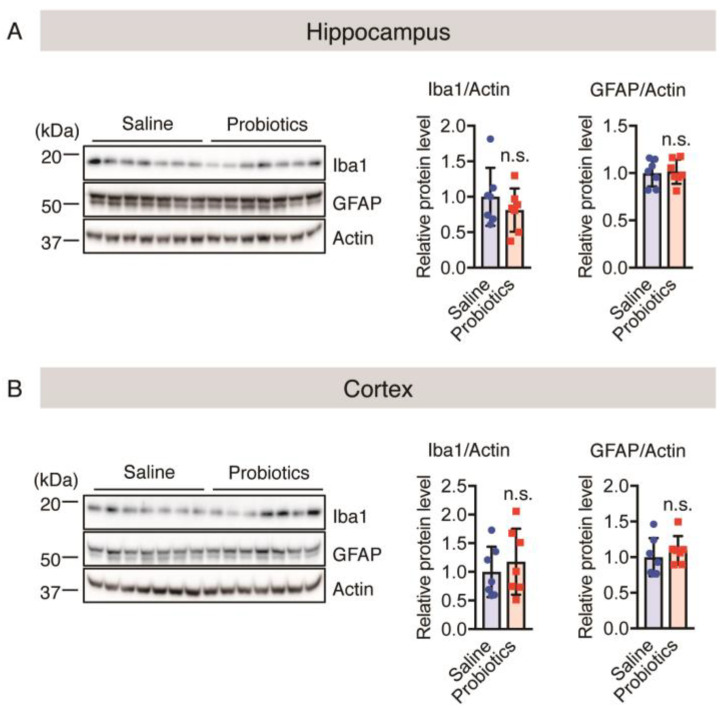
Oral *B. breve* MCC1274 administration does not change the protein levels in microglia and astrocytes in the brains of 17-month-old *App^NL-G-F^* mice. Iba1, GFAP, and actin protein levels in the hippocampus (**A**) and cortex (**B**) of aged *App^NL-G-F^* mice were assessed using western blotting post-mortem. The blots were quantified using densitometry, normalized to the level of actin protein, and expressed as relative protein levels. Data are represented as the mean ± standard deviation (*n* = 7 in each group). n.s., no significant difference; data were analyzed using Student’s *t*-test.

**Figure 6 nutrients-16-00538-f006:**
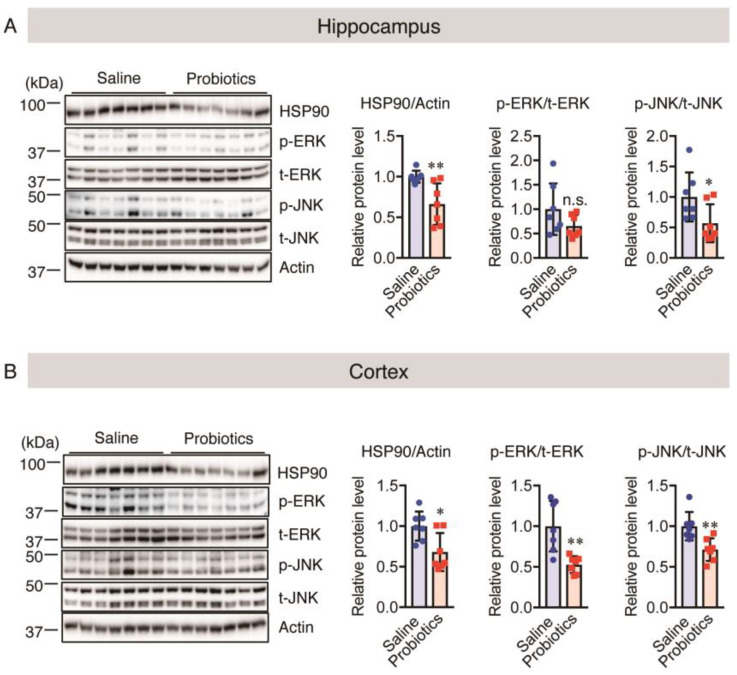
Oral *B. breve* MCC1274 administration reduces the levels of chronic stress markers in the brains of 17-month-old *App^NL-G-F^* mice. HSP90, p-ERK, t-ERK, p-JNK, t-JNK, and actin protein levels in the hippocampus (**A**) and cortex (**B**) of aged *App^NL-G-F^* mice were assessed using western blotting post-mortem. The blots were quantified using densitometry, normalized to the level of actin protein, and expressed as relative protein levels. Data are represented as the mean ± standard deviation (*n* = 7 in each group). n.s., no significant difference; * *p* < 0.05, ** *p* < 0.01, compared with the saline group; data were analyzed using Student’s *t*-test.

**Figure 7 nutrients-16-00538-f007:**
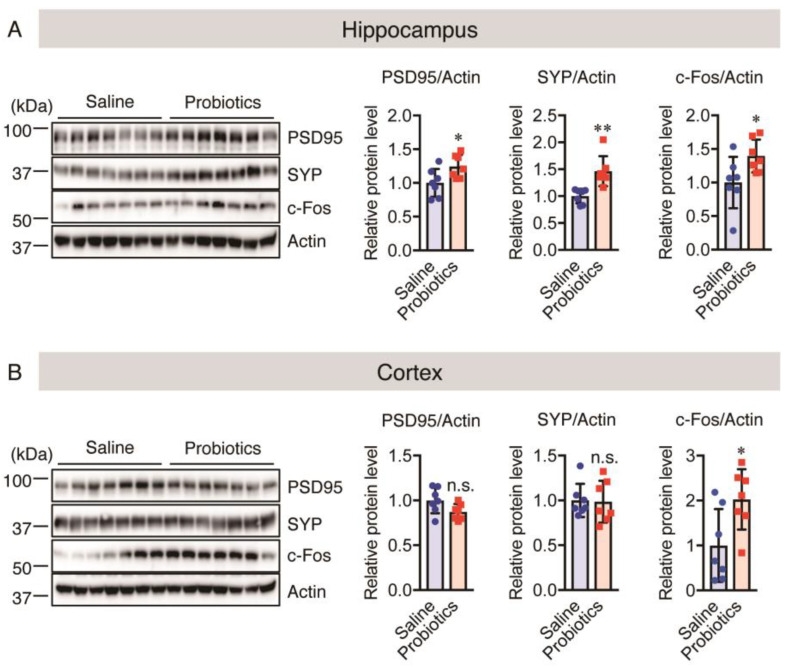
Oral *B. breve* MCC1274 administration upregulates hippocampal synaptic proteins and c-Fos protein levels in the brains of 17-month-old *App^NL-G-F^* mice. The levels of PSD95, SYP, c-Fos, NeuN, and actin proteins in the hippocampus (**A**) and cortex (**B**) of aged *App^NL-G-F^* mice were assessed using western blotting post-mortem. The blots were quantified using densitometry, normalized to the level of actin protein, and expressed as relative protein levels. Data are represented as the mean ± standard deviation (*n* = 7 in each group). n.s., no significant difference; * *p* < 0.05, ** *p* < 0.01, compared with the saline group; data were analyzed by Student’s *t*-test.

**Figure 8 nutrients-16-00538-f008:**
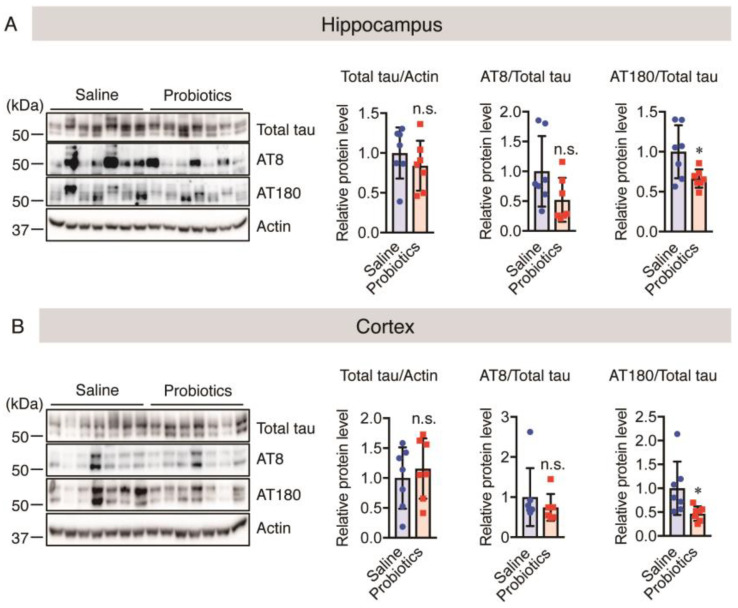
Oral *B. breve* MCC1274 administration suppresses tau hyperphosphorylation at Thr231 in the brains of 17-month-old *App^NL-G-F^* mice. Total tau level, phosphorylated tau level at sites Ser202/Thr205 (AT8) and Thr231 (AT180), and actin protein level in the hippocampus (**A**) and cortex (**B**) of aged *App^NL-G-F^* mice were assessed using western blotting post-mortem. The blots were quantified using densitometry, normalized to the level of actin protein, and expressed as relative protein levels. Data are represented as the mean ± standard deviation (*n* = 7 in each group). n.s., no significant difference; * *p* < 0.05, compared with the saline group; data were analyzed using Student’s *t*-test.

## Data Availability

All data used in this study are available from the corresponding authors upon reasonable request.

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
