# Peer review of "Potential Therapeutic Effects of Bifidobacterium breve MCC1274 on Alzheimer’s Disease Pathologies in AppNL-G-F Mice"

_nutrients, 2024, doi:10.3390/nu16040538_

Round 1

Reviewer 1 Report

Comments and Suggestions for Authors

Reviewer comments and suggestions

The authors of this study investigated the potential therapeutic effects of Bifidobacterium. breve MCC1274 probiotic on AD using 17-month-old AppNL-G-F mice with memory deficits and amyloid beta saturation in the brain. The authors reported that B. breve MCC1274 supplementation may ameliorate memory impairment via an amyloid-cascade-independent pathway. 

It decreased hippocampal and cortical levels of phosphorylated extracellular signal-regulated kinase, c-Jun N-terminal kinase, and heat shock protein 90, perhaps suppressing tau hyperphosphorylation and chronic stress. B. breve MCC1274 treatment may improve cognitive performance in rats by lowering chronic stress and tau hyperphosphorylation, resulting in increased synapse density and neuronal activity. Overall, this research points to B. breve MCC1274 as a possible treatment for AD and implies that it has anti-AD properties.

Overall, the manuscript was well written. However, a few major concerns or comments needed to be explained or modified.

  1. Line 19 Is it important to mention the previous study here in the first line of the abstract?. 
  2.  Line 58-63 Long paragraph, reduce it into two sentences, along with citing proper references
  3. Line 75-76 The line should be explained well
  4. Line 79 Some studies “Please mention the study with citation.”
  5. Line 99 I have seen recent studies on probiotics that may have enhanced the cognitive function so please cite the recent studies here and describe it well the mechanism involved for the common reader of your manuscript.
  6. Line 112-114 I think there was no need to write the conclusion or result in part at the end of the introduction section
  7. Please mention the ethical approval number in section 2.1
  8. Comments for all figures all figures were blurred so please update with the original figure
  9. Line 389-90 there was a need to mention their marker as well such as through or by...
  10. No need to mention these lines in the first paragraph of the discussion, directly mention the novelty of this study
  11. Line 458-460 is there was any possible reason for this, try to cite those studies where these levels show significant values.
  12. Line 480-481 Is it important to highlight the previous findings at several places, try to avoid it.
  13. Line 504 What was the meaning of writing objective again?
  14. The manuscript needed recent references for 2022, 23, and 24 if possible also please check references 7, 14, 15, 16, and 33.

Author Response

We would like to thank the reviewer for their excellent comments and suggestions, which have been valuable for improving the quality of our manuscript. The reviewer’s comments are addressed individually below, and we have indicated our revisions in each answer.

Comment 1) 1. Line 19 Is it important to mention the previous study here in the first line of the abstract?

Respond) We mentioned it because the idea of the current study is based on the idea of previous one and it’s a continuity for it. 

Comment 2) Line 58-63 Long paragraph, reduce it into two sentences, along with citing proper references.

Respond) The paragraph was reduced into two sentences with proper citations in line 64-68. (The bidirectional gut-brain axis communication is primarily achieved through endocrine, metabolic, and immune pathways or directly through the vagus nerve [17]. Probiotics increase the production of some neurotransmitters or activate the formation of short-chain fatty acids, or modulate the production of inflammatory cytokines [18-20]).

Comment 3) Line 75-76 The line should be explained well.

Respond) The line was explained well in line 80-84. (c-Jun N-terminal kinase (JNK) is an important kinase that activates chronic stress and may cause neurological disorders [28]. Since JNK promotes phosphorylation of glucocorticoid receptors and inhibits their transcriptional activity, thus induces glucocorticoid maladaptive influence on the brain [29,30]. As well as activation of JNK is reported during neuronal degeneration [31]).

Comment 4) Line 79 Some studies “Please mention the study with citation.”

Respond) The studies were cited in line 88.

Comment 5) Line 99 I have seen recent studies on probiotics that may have enhanced the cognitive function so please cite the recent studies here and describe it well the mechanism involved for the common reader of your manuscript. 

Respond) Thank you for your suggestion. The recent studies were cited and common mechanisms were mentioned in line 107-114. (Many recent studies have demonstrated the benefits of probiotics on cognitive function and many scientists hope to find a cure for AD. Probiotic supplementation was associated with better cognitive function in MCI older adults and benefited sleep quality in older adults [53,54]. In addition, oral administration of Bifidobacterium lactis ameliorates cognitive deficits in Aβ-injected mice through attenuation of ERK and JNK phosphorylation [55]. As well as it has been reported that Lactobacillus species act as protective from environmental stressors [56], and Bifidobacterium brevealleviated Aβ-induced cognitive impairment and enhanced synaptic function in AD mice [57]).

Comment 6) Line 112-114 I think there was no need to write the conclusion or result in part at the end of the introduction section.

Response) We deleted the conclusion at the end of introduction section.

Comment 7) Please mention the ethical approval number in section 2.1.

Response) The ethical approval number was submitted to the editor.

Comment 8) Comments for all figures all figures were blurred so please update with the original figure.

Response) This is a peer review copy and the high resolution figures will be in the final online version.  

Comment 9) Line 389-90 there was a need to mention their marker as well such as through or by...

Response) We mentioned their markers in line 390-392. (As mentioned above, B. breve MCC1274 supplementation reduced the levels of chronic stress markers such as HSP90, p-ERK, and p-JNK and improved synaptic plasticity through upregulation of PSD95 and SYP protein levels).  

Comment 10) No need to mention these lines in the first paragraph of the discussion, directly mention the novelty of this study.

Response) We deleted the first lines in the first paragraph of the discussion section.

Comment 11) Line 458-460 is there was any possible reason for this, try to cite those studies where these levels show significant values.

Response) The studies that show significant values in both Aβ and neuroinflammation were mentioned in the line 449-451. The explanation for non-changing levels of Aβ and neuroinflammation may be these levels already reached saturated levels due to aging and the progression of AD and it’s hard for the bacteria to work on it. Because in our previous studies, these levels were reduced, since the supplementation started at 3 months of age and concurrent with beginning Aβ deposition and neuroinflammation. In those previous studies, we used the same mice stain and the same probiotic, the only difference between the current study and the previous ones is the mice age. That’s why we believe that the age is the possible reason.

Comment 12) Line 480-481 Is it important to highlight the previous findings at several places, try to avoid it.

Response) Previous findings were already deleted in line 261, 406, and 482.

Comment 13) Line 504 What was the meaning of writing objective again?

Response) Objective was deleted in line 505.

Comment 14) The manuscript needed recent references for 2022, 23, and 24 if possible, also please check references 7, 14, 15, 16, and 33.

Response) Some recent references for 2023 and 2024 were added in line 45, 61, 64, 111, 112, and 114, and these references already checked.

Reviewer 2 Report

Comments and Suggestions for Authors

The potential therapeutic effects of this probiotic on AD using 17-month-old AppNL-G-F mice with memory deficits and amyloid beta saturation in the brain. B. breve MCC1274 supplementation ameliorated memory impairment via an amyloid-cascade-independent pathway. It reduced hippocampal and cortical levels of phosphorylated extracellular signal-regulated kinase and c-Jun N-terminal kinase as well as heat shock protein 90, which might have suppressed tau hyperphosphorylation and chronic stress. Moreover, B. breve MCC1274 supplementation increased hippocampal synaptic protein levels and upregulated neuronal activity. Thus, B. breve MCC1274 supplementation may alleviate cognitive dysfunction by reducing chronic stress and tau hyperphosphorylation, thereby enhancing both synaptic density and neuronal activity in 17-month-old AppNL-G-F mice. Overall, this study suggests that B. breve MCC1274 has anti-AD effects and can be used as a potential treatment for AD.

 The topic of the article is interesting. The current manuscript is well planned, prepared and the results are nicely presented. However, some parts of the article should be enhanced. Specific comments are given below;

1.     The mechanism about Alzheimer's disease should be introduced. Please refer this reference (Critical Reviews in Food Science and Nutrition, 2023, 63(29), 9816–9842.).

2.     The reference about gut microbiota and probiotic should be introduced.

3.     “2.2. Novel object recognition test”. It is critical for cognitive effect, please refer this reference (Neuroscience Letters. 2016, 631(19): 30-35.).

4.     The Clarity or Pixel need improved for the Figure.

5.     please provide whole Figure for the Western blotting.

6.     References: the authors have to check the list of references because some errors have appeared. It should be updated in recent years.

Comments on the Quality of English Language

The potential therapeutic effects of this probiotic on AD using 17-month-old AppNL-G-F mice with memory deficits and amyloid beta saturation in the brain. B. breve MCC1274 supplementation ameliorated memory impairment via an amyloid-cascade-independent pathway. It reduced hippocampal and cortical levels of phosphorylated extracellular signal-regulated kinase and c-Jun N-terminal kinase as well as heat shock protein 90, which might have suppressed tau hyperphosphorylation and chronic stress. Moreover, B. breve MCC1274 supplementation increased hippocampal synaptic protein levels and upregulated neuronal activity. Thus, B. breve MCC1274 supplementation may alleviate cognitive dysfunction by reducing chronic stress and tau hyperphosphorylation, thereby enhancing both synaptic density and neuronal activity in 17-month-old AppNL-G-F mice. Overall, this study suggests that B. breve MCC1274 has anti-AD effects and can be used as a potential treatment for AD.

 The topic of the article is interesting. The current manuscript is well planned, prepared and the results are nicely presented. However, some parts of the article should be enhanced. Specific comments are given below;

1.     The mechanism about Alzheimer's disease should be introduced. Please refer this reference (Critical Reviews in Food Science and Nutrition, 2023, 63(29), 9816–9842.).

2.     The reference about gut microbiota and probiotic should be introduced.

3.     “2.2. Novel object recognition test”. It is critical for cognitive effect, please refer this reference (Neuroscience Letters. 2016, 631(19): 30-35.).

4.     The Clarity or Pixel need improved for the Figure.

5.     please provide whole Figure for the Western blotting.

6.     References: the authors have to check the list of references because some errors have appeared. It should be updated in recent years.

Author Response

We would like to thank the reviewer for their excellent comments and suggestions, which have been valuable for improving the quality of our manuscript. The reviewer’s comments are addressed individually below, and we have indicated our revisions in each answer.

Comment 1) The mechanism about Alzheimer's disease should be introduced. Please refer this reference (Critical Reviews in Food Science and Nutrition, 2023, 63(29), 9816–9842.). 

Response) Thank you for your suggestion. The mechanism about Alzheimer's disease was added in the line 40-45. (AD pathogenesis involves pathogenic contributions from multiple components such as amyloid beta (Aβ), hyperphosphorylated tau, and glial cell. Impaired Aβ clearance and/or increased its production leads to Aβ accumulation. Also tau pathology propagates during AD progression. Aβ aggregates, together with tau accumulation can cause glial activation and the subsequent neuroinflammation [2]).

And this reference was added in line 64.

Comment 2) The reference about gut microbiota and probiotic should be introduced. 

Response) The reference was added in line 61.

Comment 3) “2.2. Novel object recognition test”. It is critical for cognitive effect, please refer this reference (Neuroscience Letters. 2016, 631(19): 30-35.).

Response) Thank you for your suggestion. We had a look to this paper and they used Morris water maze test for evaluation spatial learning and memory ability of mice and we used novel object recognition test. That’s why we couldn’t include this reference in the manuscript. 

Comment 4) The Clarity or Pixel need improved for the Figure.

Response) This is a peer review copy and the high resolution figures will be in the final online version.

Comment 5) please provide whole Figure for the Western blotting.

Response) The whole figure for the Western blotting was submitted to the editor.

Comment 6) References: the authors have to check the list of references because some errors have appeared. It should be updated in recent years.

Response) Some recent references for 2023 and 2024 were added in line 45, 61, 64, 111, 112, and 114 and these references already checked.

Round 2

Reviewer 2 Report

Comments and Suggestions for Authors

It can be accepted in the current revision.

Comments on the Quality of English Language

It can be accepted in the current revision.